# Impact of azithromycin and nitazoxanide on the enteric infections and child growth: Findings from the Early Life Interventions for Childhood Growth and Development in Tanzania (ELICIT) trial

**Godfrey Guga[1], Eric R. Houpt[2]\*, Sarah Elwood[2], Jie Liu[3], Caroline Kimathi[1], Restituta Mosha[1], Mariam Temu[1], Athanasia Maro[4], Buliga Mujaga[4], Ndealilia Swai[4], Suporn Pholwat[2], Elizabeth T. R. McQuade[5], Esto R. Mduma[1], Mark D. DeBoer[6], James Platts-Mills[2]**

1 Haydom Global Health Centre, Haydom, Tanzania, 2 University of Virginia Division of Infectious Diseases & International Health, Charlottesville, Virginia, United States of America, 3 School of Public Health, Qingdao University, Qingdao, China, 4 Biotechnology Laboratory, Kilimanjaro Clinical Research Institute, Moshi, Tanzania, 5 Department of Epidemiology, Rollins School of Public Health, Emory University, Atlanta, GA, United States of America, 6 Department of Pediatrics, University of Virginia, Charlottesville, Virginia, United States of America

\* erh6k@virginia.edu

## Abstract

### Background

Early childhood enteric infection with *Shigella/EIEC*, enteroaggregative E. coli (EAEC), *Campylobacter*, and *Giardia* has been associated with reduced child growth, yet a recent randomized trial of antimicrobial therapy to reduce these infections did not improve growth outcomes. To interrogate this discrepancy, we measured the enteric infections from this study.

### Methods

We leveraged the Early Life Interventions for Childhood Growth and Development in Tanzania (ELICIT) trial, a randomized double-blind placebo-controlled trial of antimicrobial therapy with azithromycin and nitazoxanide provided quarterly to infants from 6 to 15 months of age. We tested 5,479 stool samples at time points across the study for 34 enteropathogens using quantitative PCR.

### Results

There was substantial carriage of enteropathogens in stool. Azithromycin administration led to reductions in *Campylobacter jejuni/coli*, enteroaggregative E. coli, and *Shigella/EIEC* (absolute risk difference ranged from -0.06 to 0.24) 2 weeks after treatment however there was no effect after 3 months. There was no difference in *Giardia* after nitazoxanide administration (ARR 0.03 at the 12 month administration). When examining the effect of

**Data Availability Statement:** All relevant data are within the paper and its Supporting Information files.

**Funding:** Bill & Melinda Gates Foundation OPP1141342. Under the grant conditions of the Foundation, a Creative Commons Attribution 4.0 Generic License has already been assigned to the Author Accepted Manuscript version that might arise from this submission. The funders had no role in study design, data collection and analysis, decision to publish, or preparation of the manuscript" to the Funding Statement.

**Competing interests:** The authors have declared that no competing interests exist.

azithromycin versus placebo on the subset of children infected with specific pathogens at the time of treatment, a small increase in weight-for-age Z score was seen only in those infected with *Campylobacter jejuni/coli* (0.10 Z score, 95% CI -0.01–0.20; length-for-age Z score 0.07, 95% CI -0.06–0.20).

## Conclusion

The antimicrobial intervention of quarterly azithromycin plus or minus nitazoxanide led to only transient decreases in enteric infections with *Shigella/EIEC*, enteroaggregative E. coli (EAEC), *Campylobacter*, and *Giardia*. There was a trend towards improved growth in children infected with *Campylobacter* that received quarterly azithromycin.

## Introduction

Early enteric infections in children in low-resource settings are an established risk factor for poor growth [1]. A detailed analysis of 35,622 stool samples from infants in 8 low resource settings found that substantial decrements in length-for-age Z score at 2 years were associated with intestinal carriage of specifically *Shigella/EIEC*, enteroaggregative E. coli (EAEC), *Campylobacter*, and *Giardia*, with smaller decrements associated with *Cryptosporidium*, norovirus, typical enteropathogenic E. coli, and *Enterocytozoon bieneusi* [2]. Whether any of these associations are causal is unclear. Strategies to reduce enteric infections, including through water, sanitation and hygiene interventions, have struggled to improve child growth [3] yet have also failed to reduce bacterial and protozoal pathogen burden [4, 5].

This background formed the rationale for the Early Life Interventions for Childhood Growth and Development in Tanzania (ELICIT) trial, a randomized double-blind placebo-controlled trial of antimicrobial therapy provided to infants [6]. The intervention included azithromycin because it has activity against most *Shigella*, *Campylobacter*, and enteroaggregative E. coli. For instance, a 3 day course of azithromycin administered to Indian infants more than halved the prevalence of these 3 bacterial pathogens in stool at a 2 week time point [7]. The intervention also included nitazoxanide to provide anti-protozoal activity. For instance, nitazoxanide reduced *Giardia* cysts detected by stool microscopy in a majority of Peruvian children at a 7–10 day time point [8]. Azithromycin was provided single dose at 6, 9, 12, and 15 months of age while nitazoxanide was given as a 3-day course at 12 and 15 months (nitazoxanide is not approved until 1 year of age). Importantly, the primary outcome of ELICIT has been reported and showed no effect of the antimicrobial intervention on child length at 18 months of age [9] nor any effect on cognitive scores [10]. ELICIT was a 2 x 2 factorial design and included a nutritional intervention of nicotinamide, which could also affect enteropathogens through regulation of gut dysbiosis and secretion of antimicrobial peptides [11, 12].

Here, we examine the results of qPCR testing for 34 enteric pathogens on stool samples collected at five timepoints. Specifically, we tested stool collected at 6 months (prior to the first dose of azithromycin or placebo), 6.5 months (two weeks after receiving azithromycin or placebo), 12 months (3 months after receiving azithromycin or placebo), 12.5 months (2 weeks after receiving azithromycin and nitazoxanide or placebo) and 18 months (3 months after receiving azithromycin and nitazoxanide or placebo). This sampling time frame allowed us to assess the extent to which the antimicrobial intervention had an effect on enteric infections and to interpret the lack of effect of this antimicrobial regimen on childhood growth.

## Methods

### Study enrollment and interventions

The ELICIT study methods have been previously reported in detail [6, 9]. Enrollment occurred between September 5, 2017 and August 31, 2018. Briefly, pregnant women and mothers of newborns were approached at their homes by field team members to inform them about the study and assess interest. Inclusion criteria were maternal age ≥18 years, child age ≤14 days, and the family's stated intent to reside in the study area for the duration of the study. Exclusion criteria were multiple gestation, significant birth defect or neonatal illness, infant enrollment weight <1,500 g, and lack of intent to breastfeed. Overall 1,188 children were enrolled and allocated to one of 4 treatment groups in a 2 × 2 factorial manner in permuted blocks of eight, such that participants received either antimicrobials plus placebo, placebo plus nicotinamide, both active medications, or both placebos. Azithromycin was provided single dose (20 mg/kg) at 6, 9, 12, and 15 months of age while a 3-day course of nitazoxanide (100 mg twice daily) was provided at 12 and 15 months. The study protocol was approved by the National Institute for Medical Research (NIMR) of Tanzania (NIMR/HQ/R.8a/Vol.IX/2424 on 3 Mar 2017) and the Tanzanian Food and Drug Administration (TFDA; TFDA 0017/CTR/0005/02 on 5 July 2017)) and the Institutional Review Board at the University of Virginia (IRB-HSR #19465 on 3 May 2017). Written consent was obtained from all parents/guardians.

### Data and sample collection

Use of non-study antibiotics by study children, defined as any antibiotic that was not part of the study intervention, was identified using a standardized questionnaire at monthly visits. Child lengths using measuring boards and weights using digital scales were measured every three months. Stool samples were collected at 6, 6.5, 12, 12.5, and 18 months of age, and 1,141 children contributed at least 1 sample. The 6 and 12 month samples were required to be collected prior to administration of study antimicrobials. The study protocol was approved by the National Institute for Medical Research (NIMR) of Tanzania and the Tanzanian Food and Drug Administration (TFDA) and the Institutional Review Board at the University of Virginia. Mothers gave written informed consent to participate either during pregnancy or at the time of enrollment. The study was registered at ClinicalTrials.gov: NCT03268902.

### Stool testing

Procedures for sample extraction and testing have been previously detailed [13, 14] and the protocols are available at dx.doi.org/10.17504/protocols.io.5qpvo3k8xv4o/v1. We used custom-designed TaqMan Array Cards (ThermoFisher, Carlsbad, CA, USA) that compartmentalized probe-based quantitative PCR assays for 34 enteropathogens (S1 Table). A cycle threshold of 35 was used as the limit of detection and any detections above that were considered positive. Samples were stored at –80˚C before extraction. Bacteriophage MS2 and phocine herpesvirus were used as external controls to monitor efficiency of nucleic acid extraction and amplification. We included one extraction blank per batch and one no-template amplification control per ten cards to exclude laboratory contamination.

### Statistical analysis

Pathogens with an overall prevalence of at least 5% were included in the analyses of pathogen carriage. At a prevalence of 5% in the placebo arm, we had 80% power to detect a prevalence in the antimicrobial arm of 1.8%, a 3.2% difference; at a prevalence of 20%, we had 80% power to detect a prevalence of 13.6% in the antimicrobial arm, a 6.4% difference. To assess whether

missing pathogen data were associated with treatment arm, we fit a logistic regression model with treatment arm as the predictor, adjusted for collection time. To estimate risk differences, we used g-computation via the R package riskCommunicator with Poisson regression, with clustering by individual to account for repeated measures, to estimate the effect of treatment arm on the absolute difference in pathogen detection at each time point [15]. Poisson regression was used as an approximation of log-binomial regression [16]. Confidence intervals were estimated by bootstrap with 1000 replicates. The exposure was coded as an interaction between the antimicrobial treatment arm and the time of sample collection, and we adjusted for the treatment arm assignment for the second study intervention (nicotinamide vs. placebo). To estimate the association between exposure to non-study antibiotics in the prior month and pathogen detection, we used the same modeling approach and outcome but with an interaction between exposure to any non-study antibiotics in the prior month and sample collection timepoint, adjusting for treatment arm assignment for the antimicrobial study intervention.

To assess whether the benefit of treatment was restricted to children who had an enteric infection, we evaluated whether the effect of treatment arm on short-term changes in growth (LAZ and WAZ) differed depending on baseline detection of each pathogen in the 6- or 12-month (pretreatment) stool samples. For this analysis, we restricted to children who had complete follow-up including growth outcomes at 18 months of age. We fit a generalized linear model with a Gaussian distribution to estimate the difference in length at 9 and 15 months (three months after antibiotic treatment) between treatment arms in subgroups defined by baseline detection of each pathogen. Adjustment covariates included pretreatment length and calendar month of birth based on previously observed differences in growth by birth month [9]. Pathogens were selected for this analysis based on a previous analysis of pathogens associated with growth deficits in a multisite study that included this study site [2]. To estimate the power for this analysis, we estimated an interclass correlation coefficient of 0.05 based on our observed sample, a standard deviation of 1 and an N of 1140 in each group (given each child was included at two time points), which would have 80% power to detect a 0.269 change in LAZ or WAZ for a pathogen with a prevalence of 20% and a 0.540 change in LAZ or WAZ for a pathogen with a prevalence of 5%. As a sensitivity analysis, we also restricted to children who had stool samples available at both time points (6 and 12 months).

We then estimated the association between antimicrobial treatment arm and detection of the macrolide resistance conferring genes *mphA* and *ermB* at each timepoint using Poisson regression with generalized estimating equations and adjusting for treatment arm assignment for the nicotinamide study intervention. We used geepack [https://www.jstatsoft.org/article/view/v015i02] for these regression models with generalized estimating equations. All analyses were conducted using R version 4.2.1.

## Results

The baseline characteristics of the children in this study are summarized in Table 1. Of the 1,188 children in the ELICIT study, 1,141 contributed at least 1 stool sample for testing, of which there were 570 children in the antimicrobial arm and 571 that received placebo. Quantitative PCR testing for 34 pathogens was performed on 5,479 stool samples from collected at 6, 6.5, 12, 12.5, and 18-months. Receipt of external non-study antibiotic was common. For instance, 54–86% of children received some non-study antibiotic during the month prior to the 6, 12, and 18 month stool collections. There was no association between either treatment arm or receipt of non-study antibiotics and missing pathogen data, thus we performed complete case analyses. Length-for-age and weight-for age Z scores at 6 and 12 months were low but similar between the arms.

**Table 1. Sample testing by qPCR, receipt of non-study antibiotics, and anthropometry by intervention arm.**

| | Antimicrobial intervention | | Nicotinamide intervention | |
|---|---|---|---|---|
| | Active arm (N = 570) | Placebo arm (N = 571) | Active arm (N = 563) | Placebo arm (N = 578) |
| Samples tested by qPCR (mean, SD) | 4.79 ± 0.57 | 4.81 ± 0.65 | 4.81 ± 0.63 | 4.79 ± 0.6 |
| Children who received non-study antibiotic in prior month (n, %) | | | | |
| 6 months | 305, 54% | 329, 58% | 311, 55% | 323, 56% |
| 12 months | 414, 75% | 440, 79% | 425, 77% | 429, 77% |
| 18 months | 445, 83% | 470, 86% | 453, 85% | 462, 84% |
| Length-for-age Z score (mean, SD) | | | | |
| 6 months | -1.21 ±1.08 | -1.11 ±1.1 | -1.15 ±1.05 | -1.17 ±1.14 |
| 12 months | -1.69 ±1.01 | -1.75 ±1.03 | -1.66 ±1.02 | -1.79 ±1.02 |
| Weight-for-age Z score (mean, SD) | | | | |
| 6 months | -0.55 ±1.09 | -0.5 ±1.06 | -0.49 ±1.06 | -0.56 ±1.1 |
| 12 months | -0.72 ±0.99 | -0.82 ±0.99 | -0.75 ±0.99 | -0.79 ±0.99 |

There was substantial detection of enteric pathogens as we have noted previously [2], including of enteroaggregative *E. coli* (EAEC), typical enteropathogenic *E. coli* (EPEC), *Campylobacter*, *Shigella*, *Giardia*, *Cryptosporidium*, and *Enterocytozoon bieneusi* (Fig 1). *Shigella* and *Giardia* carriage increased substantially in the second year of life, as expected. Pathogens with an overall prevalence of at least 5% were included in further analysis.

We examined the effect of receiving study antimicrobial versus placebo on enteropathogen carriage. Fig 2 shows that receiving study antimicrobial substantially reduced detection of several pathogens in stool samples collected two weeks following the 6 or 12 month administrations. This included *Campylobacter jejuni/coli* at 6.5 and 12.5 months (absolute risk difference (ARD) -0.24; 95% CI: -0.26, -0.21 at 6.5 months and -0.28; -0.31, -0.22 at 12.5 months), tEPEC at 6.5 months (ARD -0.11; 95% CI: -0.15, -0.07), EAEC at 6.5 months (ARD -0.06; 95% CI: -0.11, - 0.02), and *Shigella/EIEC* at 12.5 months (ARD -0.16; 95% CI -0.20, -0.13). No significant difference was observed in carriage of LT-ETEC, ST-ETEC, or STEC 2 weeks after azithromycin (i.e., at the 6.5 or 12.5 month time points) nor of *Cryptosporidium* or *Giardia* 2 weeks after nitazoxanide (i.e., the 12.5 month time point). Moreover, at 3 months after antimicrobial administration (i.e., at the 12 and 18 month time points) none of the differences in *Campylobacter*, tEPEC, EAEC, or *Shigella/EIEC* carriage persisted. Stools were also tested in the children in the nicotinamide-receiving arms however no significant change in pathogens were observed in those versus placebo at any time point (S1 Fig).

Because children frequently received non-study antibiotics, we examined the effect of receiving such antibiotics on pathogens as well. Penicillins were the most frequently used class, followed by metronidazole, sulfonamides, and macrolides (S2 Fig). Use of non-study antibiotics in the month prior to stool sample collection was associated with a reduced risk of *Shigella/EIEC* (ARR −0.03; 95% CI: −0.04, −0.01) and *STEC* (ARR −0.06; 95% CI: −0.09, −0.03) at the 6 month time point, *Giardia* (ARR -0.10; 95% CI: -0.17, -0.02) at 12 months, and *E. bieneusi* (ARR −0.06; 95% CI: −0.11, −0.01) at 18 months (Fig 3).

We examined whether a growth benefit was observed with study antimicrobial in the children who carried the enteric pathogens that have most consistently been associated with growth deficits, namely *Shigella/EIEC*, enteroaggregative E. coli (EAEC), *Campylobacter jejuni/coli*, *Giardia*, and *Cryptosporidium* [2]. There were no statistically significant differences in length-for-age or weight-for-age Z scores three months after receiving study antimicrobials in infected children compared to the children not infected with these pathogens (Fig 4), but there was a marginal 0.10 (95% CI -0.01–0.20) increase in weight-for-age Z score in children

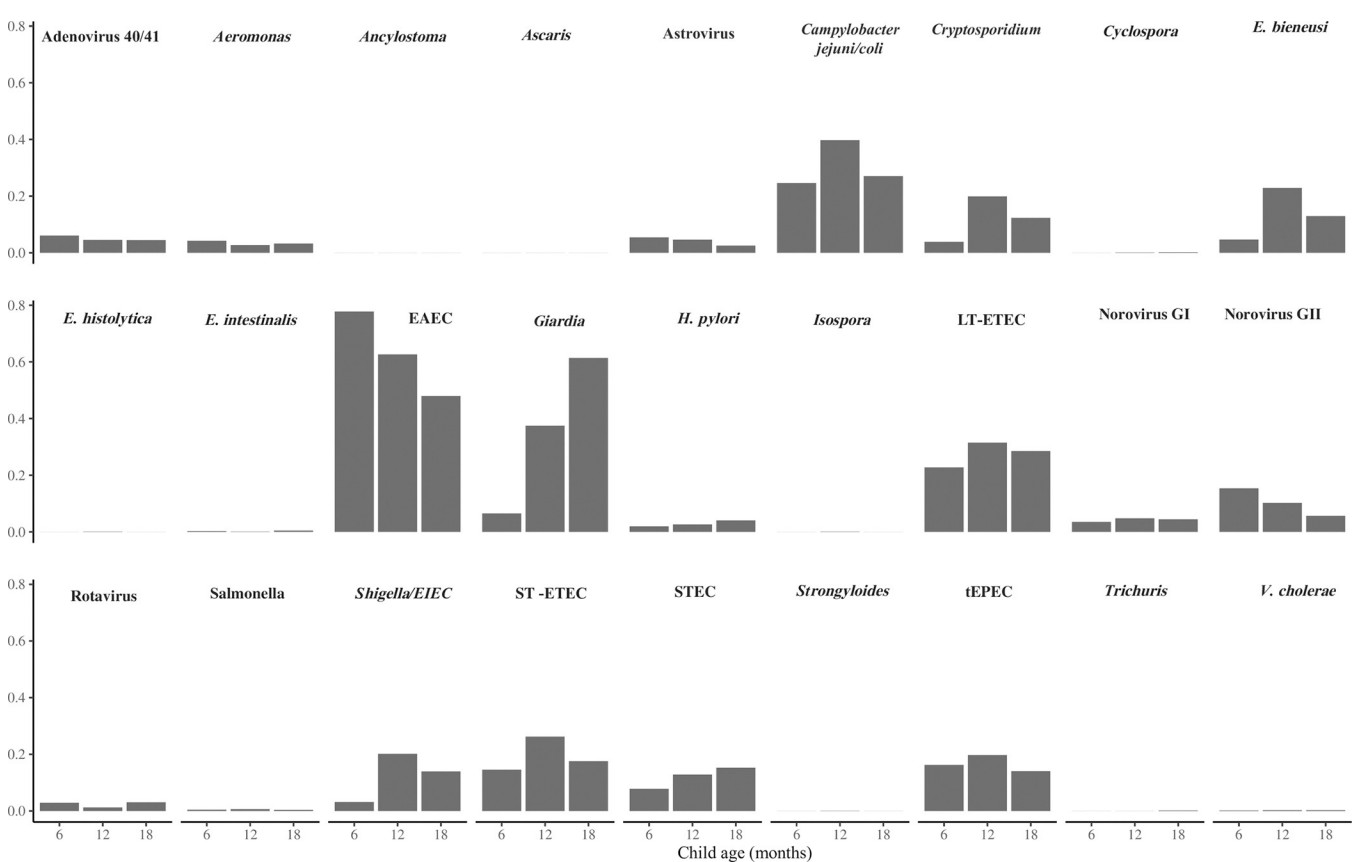

**Fig 1. Overall pathogen prevalence in the stool of ELICIT children at 6, 12, and 18 months by qPCR.** Y axis shows prevalence or proportion positive. EAEC = enteroaggregative *E. coli*; EIEC = enteroinvasive *E. coli*; LT-ETEC = heat labile toxin producing *E. coli*; ST-ETEC = heat stabile toxin producing *E. coli*; tEPEC = typical enteropathogenic *E. coli*.

infected with *Campylobacter jejuni/coli* that received antimicrobials. There was also an increase in length-for-age Z score in children with *Campylobacter jejuni/coli* that received anti-microbials however this was less statistically significant (0.07, 95% CI -0.06–0.20). The results were almost identical in a sensitivity analysis that only included children with stool PCR results at both time points.

We also tested for carriage of antimicrobial resistance genes by qPCR within this custom-ized TAC card. Receiving antimicrobials rather than placebo slightly increased the risk of detecting the macrolide resistance genes *mphA* and *ermB* two weeks after azithromycin administration at 6 months (relative risk (RR) for *mphA* = 1.11, 95% CI 1.05, 1.18; for *ermB* = 1.02, 95% CI 1.00–1.04) and 12 months (RR for *mphA* = 1.19, 95% CI 1.12, 1.26; for *ermB* = 1.03, 95% CI 1.00–1.06). These differences disappeared at 12 months (i.e., 3 months after administration).

## Discussion

The ELICIT trial administered single-dose azithromycin plus or minus nitazoxanide quarterly to infants under the hypothesis that antibiotic treatment of the intestinal infections associated with poor growth could reduce child stunting. In the primary analysis no effect on childhood growth or stunting was seen at 18 months of age [9]. The present work analyzed 5,479 stool samples in order to better interpret the findings and assess the hypothesis.

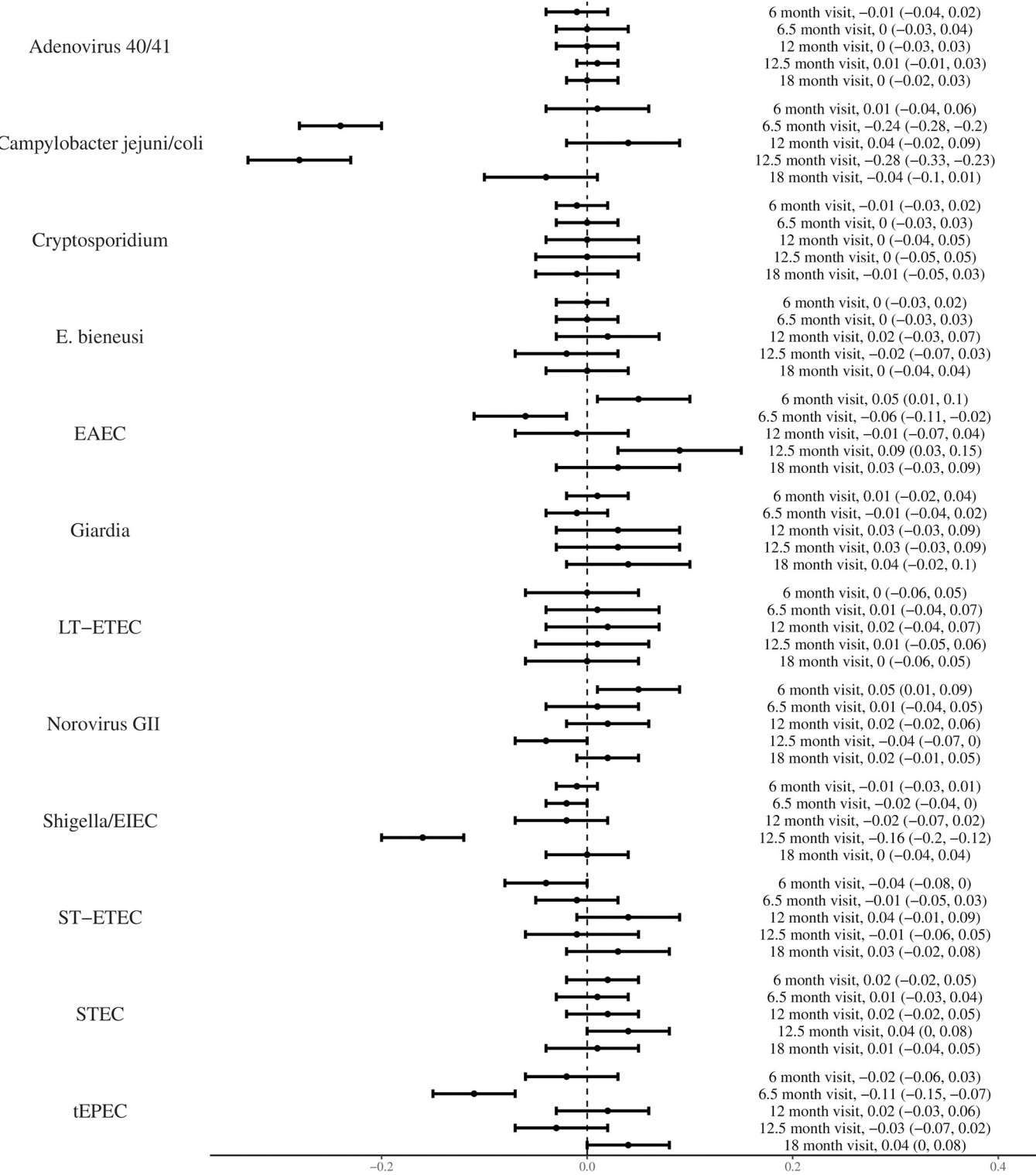

**Fig 2. Difference in enteropathogen detection in the antibiotic versus placebo arms at 6, 6.5, 12, 12.5, and 18 months.** X axis is the absolute difference in pathogen prevalence between the antibiotic and placebo arms. Negative values to the left of 0.0 represent lower carriage in the active arm.

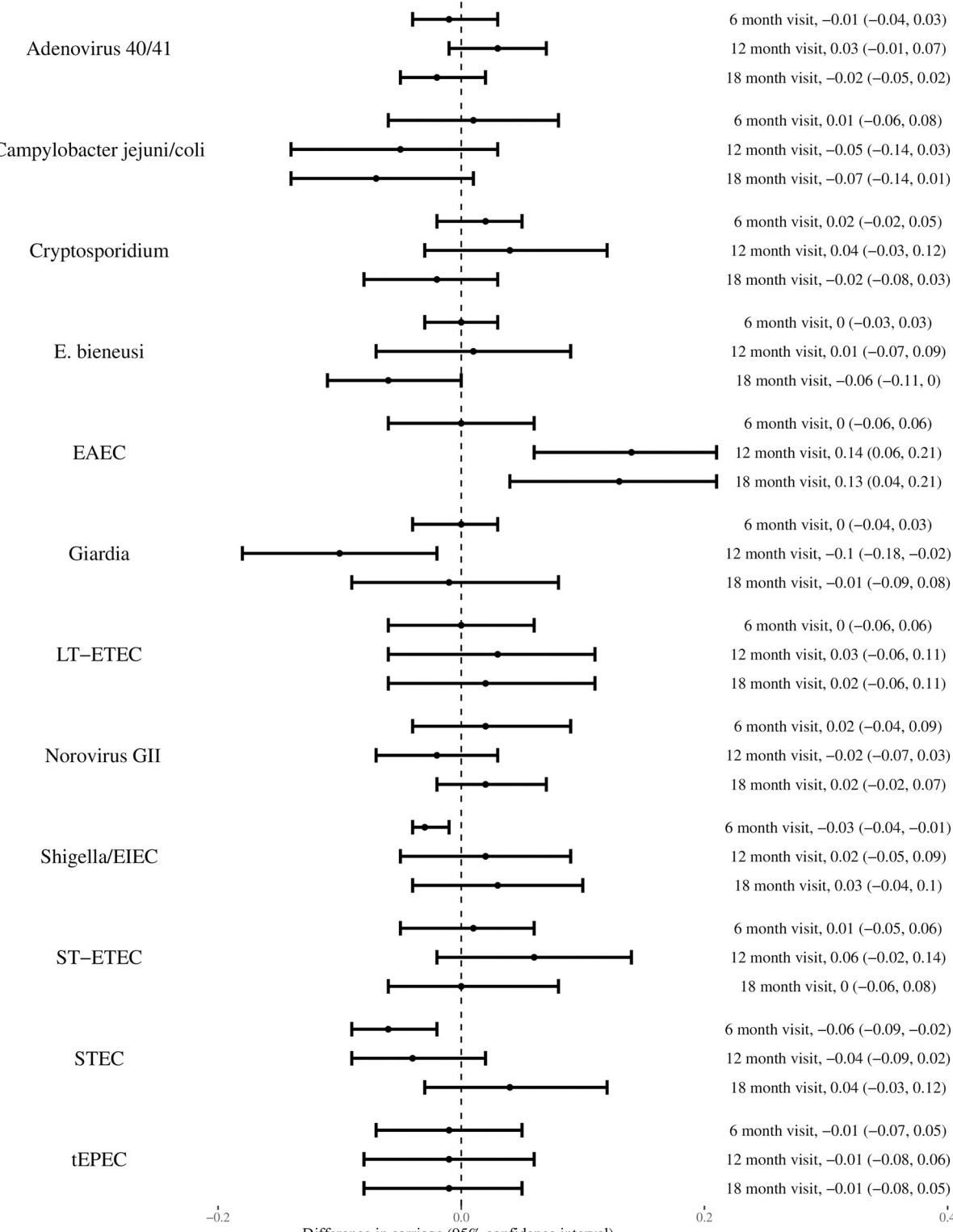

**Fig 3. Difference in pathogen detection in children that recently received non-study antibiotics.** X axis is the absolute difference in those that received non-study antibiotics during the month prior to 6, 12, and 18 months versus those that did not receive non-study antibiotics. Negative values to the left of 0.0 represent lower carriage with receipt of non-study antibiotics.

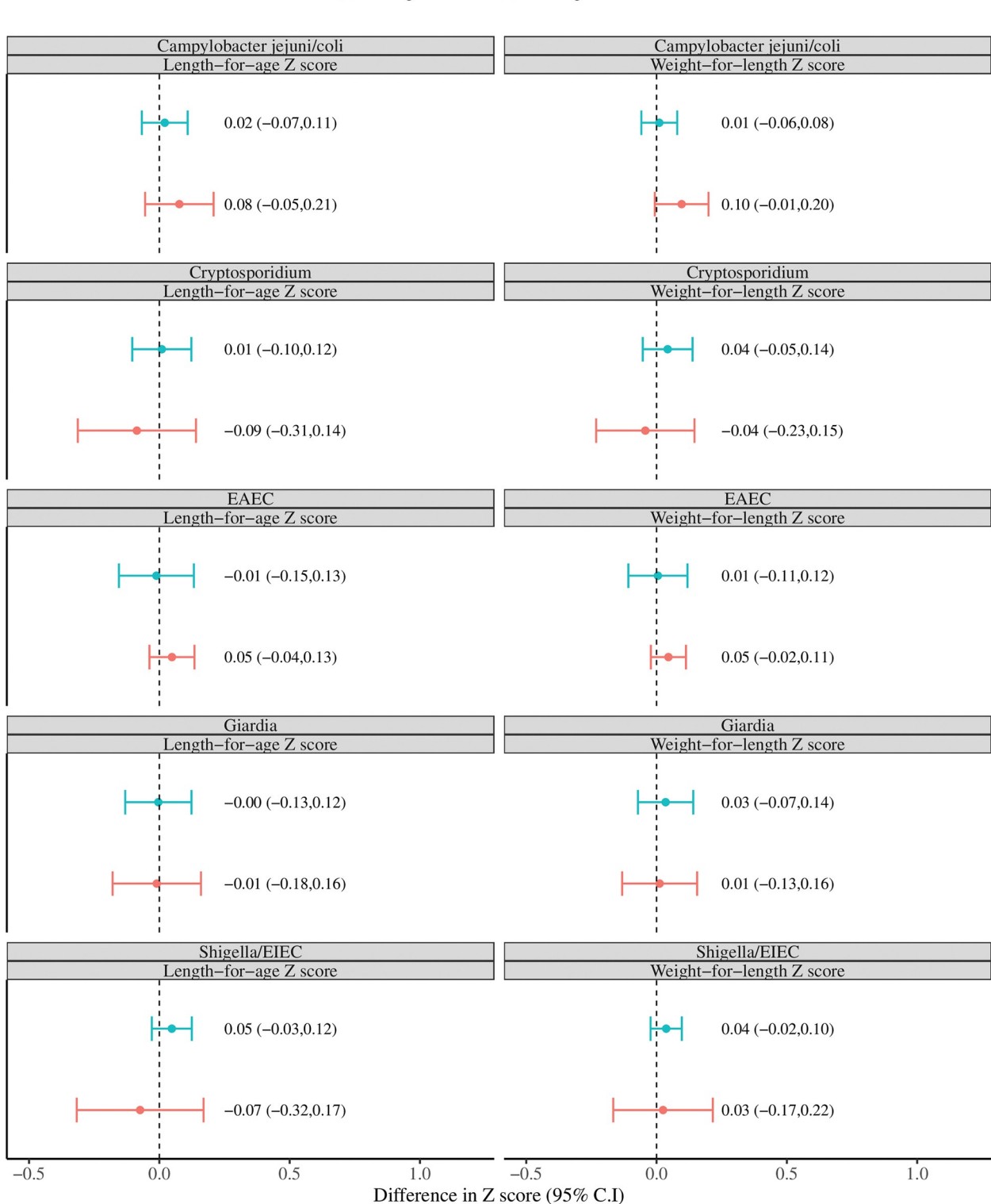

**Fig 4. Difference in 3 month growth outcomes by treatment arm stratified by pathogen detection at 6 and 12 months of age.** Shown are the subset of children at 6 and 12 months with *Shigella/EIEC*, enteroaggregative E. coli (EAEC), *Campylobacter*, *Giardia*, or *Cryptosporidium* detected compared with those without the pathogen detected. The X axis is the difference in the length (left) or weight (right) for age Z score in the antimicrobial arm–the Z score in the placebo arm, such that a difference in Z score above 0.0 represents an increase in child length or weight with antimicrobials.

First, we did observe a clear and large effect of azithromycin on *Campylobacter jejuni/coli* and *Shigella/EIEC*, with an absolute difference in *Shigella/EIEC* carriage of about ~16% and *Campylobacter jejuni/coli* of about ~24–28% of children two weeks after administration. There was also some clearance seen of EAEC and tEPEC albeit of smaller magnitude. In an earlier study of Indian infants treated for 3 days with azithromycin we found greater reductions in EAEC, EPEC, *Campylobacter*, and *Shigella/EIEC* of about 50% [7] Prior work from Niger has also shown that *Shigella* was reduced by azithromycin mass drug administration to a greater extent than seen here [17]. Therefore, while reductions in pathogens were observed with azithromycin, it was lower than expected, particularly for diarrheagenic *E. coli*.

It was surprising to not see an effect of azithromycin against ETEC, as this antibiotic clearly has shown an effect on ETEC diarrhea [18]. It was also surprising that that there was no effect ascribable to nitazoxanide against *Giardia* or *Cryptosporidium*. It is possible that the high use of external non-study antimicrobials in the community obscured the study's antimicrobial effects. For instance, the two most common non-study antibiotics were penicillins and metronidazole, and these could have "pre-reduced" *Shigella/EIEC* and *Giardia*, respectively, such that additional study azithromycin or nitazoxanide had a limited ability to show impact. It is also possible than antimicrobial resistance to azithromycin played some role, as rates of resistance to fecal E. coli in this study were previously noted to be 23–32% [9]. Perhaps the most important explanation for the low efficacy of study antimicrobials, however, was the high force of re-infection with these enteric infections from the environment. Notably, at 3 months after antimicrobials all of the earlier reductions were lost.

Therefore the underlying hypothesis of the ELICIT study was not able to be completely tested, as it remains possible that a better antimicrobial regimen that better and more durably reduces the pathogens associated with poor growth (*Shigella/EIEC*, EAEC, *Giardia*, and *Campylobacter*) could reduce stunting. In this setting however, it appears that this would require frequent dosing of azithromycin and is probably impractical.

There was a suggestion of improved weight-for-age Z score and length-for-age Z score in the subset of children infected with *Campylobacter jejuni/coli* that received azithromycin. It is certainly plausible that certain pathogens associated with poor growth are more important or causal (such as *Campylobacter*) than others, and if *Campylobacter* could be selectively targeted this would be important for further study. However the observed effect sizes were small ($< =$ 0.10 Z score) and thus even if true may not translate into significant improvements in child health. Of course it is possible that more complete removal of this or other pathogens from the gut could have greater effects.

Limitations of this work include that our sampling frame only tested stools at 6 monthly time points so we do not know the kinetics of how long the azithromycin effect lasted or how quickly reinfections occurred. Additionally, the PCR approach could detect prolonged shedding of pathogens killed after antibiotics which could also obscure an antimicrobial effect.

In sum, the antimicrobial regimen tested in this study was not as effective as expected in treating the candidate enteropathogens. We do not believe that periodic azithromycin mass drug administration is the proper tool to improve childhood growth, at least in this setting, since the aggressive regimens of antimicrobial required to reduce enteric infections would carry complex risks and antimicrobial resistance. The suggestion that reducing *Campylobacter* could improve childhood growth should be pursued in future studies. If true, since *Campylobacter* is such a highly prevalent enteric pathogen, an effective and narrow-spectrum method to reduce this infection could be beneficial and should be studied given the intractable problem of childhood stunting.

## Supporting information

**S1 Table. Pathogens and gene targets tested by PCR in this study.**
(DOCX)

**S2 Table. Complete ELICIT TAC data.**
(CSV)

**S3 Table. ELICIT TAC data dictionary.**
(CSV)

**S1 Fig. Difference in pathogen detection in the nicotinamide versus placebo arms at 6, 6.5, 12, 12.5, and 18 months.** X axis is the absolute reduction in pathogen prevalence between the nicotinamide–placebo. Negative values to the left of 0.0 represent reduction with nicotinamide. 95% confidence intervals are included.
(DOCX)

**S2 Fig. Prevalence of non-study antibiotic use by drug class and month.**
(DOCX)

## Author Contributions

**Conceptualization:** Eric R. Houpt, Esto R. Mduma, Mark D. DeBoer, James Platts-Mills.

**Data curation:** Sarah Elwood, Jie Liu, Elizabeth T. R. McQuade, James Platts-Mills.

**Formal analysis:** Godfrey Guga, Sarah Elwood, Jie Liu, Elizabeth T. R. McQuade, Mark D. DeBoer, James Platts-Mills.

**Funding acquisition:** Esto R. Mduma.

**Investigation:** Eric R. Houpt, Caroline Kimathi, Restituta Mosha, Mariam Temu, Athanasia Maro, Buliga Mujaga, Ndealilia Swai, Esto R. Mduma, Mark D. DeBoer, James Platts-Mills.

**Methodology:** Godfrey Guga, Eric R. Houpt, Jie Liu, Caroline Kimathi, Restituta Mosha, Mariam Temu, Athanasia Maro, Buliga Mujaga, Ndealilia Swai, Suporn Pholwat, Mark D. DeBoer, James Platts-Mills.

**Project administration:** Esto R. Mduma, Mark D. DeBoer, James Platts-Mills.

**Supervision:** Godfrey Guga, Eric R. Houpt, Jie Liu, Elizabeth T. R. McQuade, Esto R. Mduma, Mark D. DeBoer, James Platts-Mills.

**Writing – original draft:** Eric R. Houpt, James Platts-Mills.

**Writing – review & editing:** Godfrey Guga, Eric R. Houpt, Sarah Elwood, Jie Liu, Caroline Kimathi, Restituta Mosha, Mariam Temu, Athanasia Maro, Buliga Mujaga, Ndealilia Swai, Suporn Pholwat, Elizabeth T. R. McQuade, Esto R. Mduma, Mark D. DeBoer.

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
