## [Decision Letter · Decision Letter 0]

29 Aug 2023

PONE-D-23-20623Impact of azithromycin and nitazoxanide on the enteric infections and child growth: findings from the Early Life Interventions for Childhood Growth and Development in Tanzania (ELICIT) trialPLOS ONE

Dear Dr. Houpt,

Thank you for submitting your manuscript to PLOS ONE. After careful consideration, we feel that it has merit but does not fully meet PLOS ONE’s publication criteria as it currently stands. Therefore, we invite you to submit a revised version of the manuscript that addresses the points raised during the review process.

 Please pay particular attention to Reviewer #1's comments regarding sample size and the impact on the conclusions of the study.

We look forward to receiving your revised manuscript.

Kind regards,

David Joseph Diemert, M.D.

Academic Editor

PLOS ONE

“Funding: Bill & Melinda Gates Foundation OPP1141342.  Under the grant conditions of the Foundation, a Creative Commons Attribution 4.0 Generic License has already been assigned to the Author Accepted Manuscript version that might arise from this submission.  The funders had no role in study design, data collection and analysis, decision to publish, or preparation of the manuscript.”

 “NO”

Please include your amended statements within your cover letter; we will change the online submission form on your behalf."

Reviewers' comments:

Reviewer's Responses to Questions

**Comments to the Author**

1. Is the manuscript technically sound, and do the data support the conclusions?

Reviewer #1: Partly

Reviewer #2: Yes

2. Has the statistical analysis been performed appropriately and rigorously? 

Reviewer #1: No

Reviewer #2: Yes

3. Have the authors made all data underlying the findings in their manuscript fully available?

Reviewer #1: Yes

Reviewer #2: Yes

4. Is the manuscript presented in an intelligible fashion and written in standard English?

Reviewer #1: Yes

Reviewer #2: Yes

5. Review Comments to the Author

Reviewer #1: My main concern is that the original sample size was not calculated for this study. As a result, the non-significance of the findings is inconclusive, as we cannot determine whether it is due to the actual effect being absent or simply due to a lack of statistical power. The authors may consider conducting a sensitivity power analysis.

Furthermore, it is essential to adjust p-values when conducting multiple simultaneous tests to avoid potential false positives. In the discussion of the results, it is important to consider both statistical significance and clinical significance.

Regarding the primary analysis, it is important to clarify whether the Poisson regression model will account for repeated measures properly. How were the Poisson or normal distributions selected?

Dealing with missing values and any instances of lost to follow up is critical. The approach to handling missing data should be clearly outlined in the methodology section. Proper techniques such as imputation or sensitivity analyses should be employed to minimize biases that missing data may introduce.

Minor concerns:

“Non study” meaning “non prescription”?

Line 141 better say "no significant difference" than "no difference".

Line 95 reduction meaning change from baseline?

Line 97 treatment will be the primary factor, not an adjustment.

Need more details of G-computation for general audiences.

Line 114 R version is not correct. And cite of geepack is missing.

Table 1 title of “baseline characteristics” is not correct.

Reviewer #2: This is a well written secondary analysis of the results of the ELCIIT trial in Tanzania, investigating possible underlying causes explaining the failure to detect an effect on child growth by repeated administration of antimicrobials. The authors have tested stool samples from treated children and controls for a panel of pathogenic microorganisms and found that while the antimicrobials had a short term effect on some pathogens, the effect was not sustained, possibly because of a high force of infection. Treatment with non-study antimicrobials may have partly masked the effect of the study antimicrobials.

These findings are valuable in understanding both the results of this specific trial and moreover the complex dynamics of pathogen exposure and infection in low- and middle-income countries.

The paper is well written with clearly laid out tales and figures.

My main question is for the authors to add some discussion of the power of the current analysis. The study has been powered to measure the effects of the primary objectives, and power of the current analysis may have been reduced. This information would help interpret the wide confidence intervals in many effects. It looks like the authors themselves are also somewhat ambiguous in their interpretations. In the abstract and results, they mention the effects of treatment on LAZ and WAZ in children who carried Campylobacter. While the effects on WAZ are marginally significant at alpha = 5%, the results on LAZ are not. This difference is not clearly stated. Moreover, in the discussion (lines 201-205), the results with Campylobacter are highlighted as potentially worth further investigation. This argument would be substantiated by a low power of the study.

A minor comment is that in line 114, a reference is missing.

6. PLOS authors have the option to publish the peer review history of their article (what does this mean?). If published, this will include your full peer review and any attached files.

Reviewer #1: No

Reviewer #2: **Yes: **Arie Havelaar, University of Florida

---

## [Author Response · Author response to Decision Letter 0]

9 Oct 2023

Editor comments:

We recommend that you deposit your laboratory protocols in protocols.io to enhance the reproducibility of your results.

Done, and listed in the methods. dx.doi.org/10.17504/protocols.io.5qpvo3k8xv4o/v1

We have moved the “Funding: Bill & Melinda Gates Foundation OPP1141342. Under the grant conditions of the Foundation, a Creative Commons Attribution 4.0 Generic License has already been assigned to the Author Accepted Manuscript version that might arise from this submission. The funders had no role in study design, data collection and analysis, decision to publish, or preparation of the manuscript” to the Funding Statement.

Reviewers' comments:

Reviewer #1: My main concern is that the original sample size was not calculated for this study. As a result, the non-significance of the findings is inconclusive, as we cannot determine whether it is due to the actual effect being absent or simply due to a lack of statistical power. The authors may consider conducting a sensitivity power analysis.

The ELICIT trial was powered for the effect of the interventions on the primary outcome of growth; this was a secondary analysis. We have added a post-hoc power calculation for the primary question in this analysis, namely whether antimicrobial treatment caused a reduction in pathogen carriage. Specifically, “Pathogens with an overall prevalence of at least 5% were included in the analyses of pathogen carriage. At a prevalence of 5% in the placebo arm, we had 80% power to detect a prevalence in the antimicrobial arm of 1.8%, a 64% relative reduction; at a prevalence of 20%, we had 80% power to detect a prevalence of 13.6% in the antimicrobial arm, a 32% relative reduction.” We have also added a post-hoc power calculation for the effect of antimicrobial arm on child growth in the subset of children with specific pathogens detected at 6 and 12 months, namely “To estimate the power for this analysis, we used a T test assuming a standard deviation of 1 and N of 1140 in each group (given each child was included at two time points), which would have 80% power to detect a 0.263 change in LAZ or WAZ for a pathogen with a prevalence of 20% and a 0.529 change in LAZ or WAZ for a pathogen with a prevalence of 5%.”

Furthermore, it is essential to adjust p-values when conducting multiple simultaneous tests to avoid potential false positives. In the discussion of the results, it is important to consider both statistical significance and clinical significance.

The manuscript does not include any P values, however we agree with the reviewer that the clinical significance of the findings is important to consider. For the primary analysis of the effect of antimicrobial arm on pathogen detection, we have reported risk differences to make the clinical significance clear. The observed differences for Campylobacter and Shigella are very large. We discuss the effect sizes more clearly now in the discussion. We also have expanded our discussion of the effect sizes for child growth, namely “The observed effect sizes were small (<= 0.10 Z score) and thus even if true may not translate into significant improvements in child health, however it is possible that more comprehensive removal of this pathogen from the early-child gut would translate to more significant improvements. This analysis was also underpowered for growth improvements for less prevalent pathogens.”

Regarding the primary analysis, it is important to clarify whether the Poisson regression model will account for repeated measures properly. How were the Poisson or normal distributions selected?

Poisson regression was used as an approximation of log-binomial regression. For the analyses of the effect of study and non-study antibiotics on pathogen carriage, this was used in concert with g-computation, with clustering by individual to account for repeated measures. This has been clarified in the methods and a citation has been added.

Dealing with missing values and any instances of lost to follow up is critical. The approach to handling missing data should be clearly outlined in the methodology section. Proper techniques such as imputation or sensitivity analyses should be employed to minimize biases that missing data may introduce.

We agree that this should be clarified. For the primary analysis of the effect of antimicrobial arm on pathogen carriage, we have added to the Methods, “to assess whether missing pathogen data were associated with treatment arm, we fit a logistic regression model with treatment arm as the predictor, adjusted for collection time” and in the Results, “there was no association between either treatment arm or receipt of non-study antibiotics and missing pathogen data, thus we performed complete case analyses.” For the analysis of the effect of antimicrobial arm on growth in the subset of children who had pathogen carriage, we have clarified that this analysis only included children with growth outcomes available. We also added a sensitivity analysis in which we limited only to children who had stool samples available at both time points (6 and 12 months). This did not change the results in a meaningful way, and this is stated in the Results text.

Minor concerns:

“Non study” meaning “non prescription”? We have clarified that “non-study” means any antibiotic that was not part of the study intervention.

Line 141 better say "no significant difference" than "no difference". 

We have made the suggested edit.

Line 95 reduction meaning change from baseline? 

We agree that this was unclear. We have changed “reduction” to “difference” here and have preferred “difference” when discussion the estimated effects throughout, but have inferred “reduction” at some points in the discussion based on these findings.

Line 97 treatment will be the primary factor, not an adjustment. 

The “treatment” referenced in Line 97 is the second (factorial) study intervention, not antimicrobial arm, thus we believe that this is correct as written.

Need more details of G-computation for general audiences. 

We have added additional detail and a citation that describes the RiskCommunicator R package that was used.

Line 114 R version is not correct. And cite of geepack is missing. 

We have corrected the R version and added a citation for geepack.

Table 1 title of “baseline characteristics” is not correct. 

We agree with the reviewer and have renamed the Table 1 title.

Reviewer #2: This is a well written secondary analysis of the results of the ELCIIT trial in Tanzania, investigating possible underlying causes explaining the failure to detect an effect on child growth by repeated administration of antimicrobials. The authors have tested stool samples from treated children and controls for a panel of pathogenic microorganisms and found that while the antimicrobials had a short term effect on some pathogens, the effect was not sustained, possibly because of a high force of infection. Treatment with non-study antimicrobials may have partly masked the effect of the study antimicrobials.

These findings are valuable in understanding both the results of this specific trial and moreover the complex dynamics of pathogen exposure and infection in low- and middle-income countries. The paper is well written with clearly laid out tales and figures.

We appreciate the reviewer’s comments as we believe that this is a unique opportunity to understand pathogen exposure and carriage in an interventional trial.

My main question is for the authors to add some discussion of the power of the current analysis. The study has been powered to measure the effects of the primary objectives, and power of the current analysis may have been reduced. This information would help interpret the wide confidence intervals in many effects. It looks like the authors themselves are also somewhat ambiguous in their interpretations. In the abstract and results, they mention the effects of treatment on LAZ and WAZ in children who carried Campylobacter. While the effects on WAZ are marginally significant at alpha = 5%, the results on LAZ are not. This difference is not clearly stated. Moreover, in the discussion (lines 201-205), the results with Campylobacter are highlighted as potentially worth further investigation. This argument would be substantiated by a low power of the study.

Please also see the responses to Reviewer #1. The ELICIT trial was powered for the effect of each factorial intervention on child growth. Here, we used all available samples from all children, thus maximizing the power to understand the effect of the antimicrobial intervention on pathogen carriage. However, we have added post-hoc power calculations, as described above, to provide some context.

Finally, we modified the discussion to be more guarded in the interpretations, namely:

… the observed effect sizes were small (<= 0.10 Z score) and thus even if true may not translate into significant improvements in child health. Of course it is possible that more complete removal of this or other pathogens from the gut could have greater effects

A minor comment is that in line 114, a reference is missing. 

We have added the appropriate reference.

---

## [Decision Letter · Decision Letter 1]

25 Oct 2023

Impact of azithromycin and nitazoxanide on the enteric infections and child growth: findings from the Early Life Interventions for Childhood Growth and Development in Tanzania (ELICIT) trial

PONE-D-23-20623R1

Dear Dr. Houpt,

We’re pleased to inform you that your manuscript has been judged scientifically suitable for publication and will be formally accepted for publication once it meets all outstanding technical requirements.

Kind regards,

David Joseph Diemert, M.D.

Academic Editor

PLOS ONE

Additional Editor Comments (optional):

Reviewers' comments:

Reviewer's Responses to Questions

**Comments to the Author**

1. If the authors have adequately addressed your comments raised in a previous round of review and you feel that this manuscript is now acceptable for publication, you may indicate that here to bypass the “Comments to the Author” section, enter your conflict of interest statement in the “Confidential to Editor” section, and submit your "Accept" recommendation.

Reviewer #1: All comments have been addressed

Reviewer #2: All comments have been addressed

2. Is the manuscript technically sound, and do the data support the conclusions?

Reviewer #1: (No Response)

Reviewer #2: Yes

3. Has the statistical analysis been performed appropriately and rigorously? 

Reviewer #1: (No Response)

Reviewer #2: Yes

4. Have the authors made all data underlying the findings in their manuscript fully available?

Reviewer #1: (No Response)

Reviewer #2: Yes

5. Is the manuscript presented in an intelligible fashion and written in standard English?

Reviewer #1: (No Response)

Reviewer #2: Yes

6. Review Comments to the Author

Reviewer #1: (No Response)

Reviewer #2: (No Response)

7. PLOS authors have the option to publish the peer review history of their article (what does this mean?). If published, this will include your full peer review and any attached files.

Reviewer #1: No

Reviewer #2: **Yes: **Arie H. Havelaar

---

## [Editor Report · Acceptance letter]

14 Nov 2023

PONE-D-23-20623R1 

Impact of azithromycin and nitazoxanide on the enteric infections and child growth: findings from the Early Life Interventions for Childhood Growth and Development in Tanzania (ELICIT) trial 

Dear Dr. Houpt:

I'm pleased to inform you that your manuscript has been deemed suitable for publication in PLOS ONE. Congratulations! Your manuscript is now with our production department. 

Kind regards, 

on behalf of

Dr. David Joseph Diemert 

Academic Editor

PLOS ONE